# In-House Immunofluorescence Assay for Detection of SARS-CoV-2 Antigens in Cells from Nasopharyngeal Swabs as a Diagnostic Method for COVID-19

**DOI:** 10.3390/diagnostics11122346

**Published:** 2021-12-13

**Authors:** Athene Hoi-Ying Lam, Jian-Piao Cai, Ka-Yi Leung, Ricky-Ruiqi Zhang, Danlei Liu, Yujing Fan, Anthony Raymond Tam, Vincent Chi-Chung Cheng, Kelvin Kai-Wang To, Kwok-Yung Yuen, Ivan Fan-Ngai Hung, Kwok-Hung Chan

**Affiliations:** 1Department of Medicine, Li Ka Shing Faculty of Medicine, University of Hong Kong, Hong Kong, China; athenelx@connect.hku.hk (A.H.-Y.L.); zhangrq@hku.hk (R.-R.Z.); liudl@hku.hk (D.L.); jyfan@connect.hku.hk (Y.F.); 2Department of Microbiology, Li Ka Shing Faculty of Medicine, University of Hong Kong, Hong Kong, China; caijuice@hku.hk.com (J.-P.C.); joy2ky@hku.hk (K.-Y.L.); kelvinto@hku.hk (K.K.-W.T.); kyyuen@hku.hk (K.-Y.Y.); 3Department of Medicine, Queen Mary Hospital, Hong Kong, China; antamwf@connect.hku.hk; 4Department of Microbiology, Queen Mary Hospital, Hospital Authority, Hong Kong, China; vcccheng@hku.hk; 5State Key Laboratory for Emerging Infectious Diseases, Li Ka Shing Faculty of Medicine, University of Hong Kong, Hong Kong, China; 6Carol Yu Centre for Infection, Li Ka Shing Faculty of Medicine, University of Hong Kong, Hong Kong, China

**Keywords:** immunofluorescence assay, detection of SARS-CoV-2, nasopharyngeal swabs, COVID-19

## Abstract

Immunofluorescence is a traditional diagnostic method for respiratory viruses, allowing rapid, simple and accurate diagnosis, with specific benefits of direct visualization of antigens-of-interest and quality assessment. This study aims to evaluate the potential of indirect immunofluorescence as an in-house diagnostic method for SARS-CoV-2 antigens from nasopharyngeal swabs (NPS). Three primary antibodies raised from mice were used for immunofluorescence staining, including monoclonal antibody against SARS-CoV nucleocapsid protein, and polyclonal antibodies against SARS-CoV-2 nucleocapsid protein and receptor-binding domain of SARS-CoV-2 spike protein. Smears of cells from NPS of 29 COVID-19 patients and 20 non-infected individuals, and cells from viral culture were stained by the three antibodies. Immunofluorescence microscopy was used to identify respiratory epithelial cells with positive signals. Polyclonal antibody against SARS-CoV-2 N protein had the highest sensitivity and specificity among the three antibodies tested, detecting 17 out of 29 RT-PCR-confirmed COVID-19 cases and demonstrating no cross-reactivity with other tested viruses except SARS-CoV. Detection of virus-infected cells targeting SARS-CoV-2 N protein allow identification of infected individuals, although accuracy is limited by sample quality and number of respiratory epithelial cells. The potential of immunofluorescence as a simple diagnostic method was demonstrated, which could be applied by incorporating antibodies targeting SARS-CoV-2 into multiplex immunofluorescence panels used clinically, such as for respiratory viruses, thus allowing additional routine testing for diagnosis and surveillance of SARS-CoV-2 even after the epidemic has ended with low prevalence of COVID-19.

## 1. Introduction

An integrated approach with considerations of clinical, radiological, epidemiological and molecular evidence is employed for diagnosis of coronavirus disease 19 (COVID-19) caused by severe acute respiratory syndrome coronavirus 2 (SARS-CoV-2), [1]. Early detection for isolation, contact tracing and treatment of cases is essential to prevent further spread of disease to uninfected population, as well as permanent physical damage and long-term complications of infected patients due to delayed treatment [2]. Different diagnostic methods have been developed, including nucleic acid-based, whole virion-based, antibody-based and antigen-based tests. Reverse transcription polymerase chain reaction (RT-PCR) is the current gold standard for SARS-CoV-2 detection [3], which is highly sensitive but requires designated thermocyclers and specific expertise in molecular techniques, high expenses and generally long turnaround time due to complicated procedures, although sample-to-result PCR-based methods have been developed for use in non-specialized laboratories to simplify and shorten the process with the cost per reaction being high. Other nucleic acid detection methods, such as sequencing and droplet digital PCR [4], harbor similar disadvantages in terms of time, cost and complexity. Direct viral isolation by electron microscopy or culture in cell lines, such as Vero E6 monkey kidney cells, could be used as a diagnostic tool, revealing cytopathic effects and presence of live SARS-CoV-2 that indicates infectivity of clinical samples; nevertheless, its complicated procedures that involve handling of dangerous live viruses in biosafety level 3 laboratories for a long duration, and its insensitivity prevent its application as a routine testing method [5,6,7]. Seroconversion enabling detection of antibodies occurs days after initial infection, with serological tests targeting IgA, IgM and IgG being developed for identifying any past or current infections and studying seroprevalence [8,9,10]. Antigen detection assays, such as ELISA, were developed for identifying specific structural proteins of SARS-CoV-2 [11], and various lateral-flow rapid antigen tests utilizing immunochromatographic assay or fluorescence immunoassay are widely used as point-of-care tests given their simplicity, cost-effectiveness and low technical requirements, although accuracy and utility of rapid antigen tests have been reported to be variable [12,13]. Immunofluorescence have been established as a potential antibody detection method for SARS-CoV-2 antibodies [14], but its reversed application to detect specific viral antigens has not been studied. 

Antigen-based detection methods rely on the identification of structural proteins of coronaviruses, that are essential in maintaining their viral cycle. Spike (S) proteins are transmembrane proteins protruding from a viral envelope for interactions with the environment, such as entry into host cells [15], in which S1 subunit harbors the receptor-binding domain (RBD) and mediates recognition and binding of virus to receptors that allows viral attachment to host cells [16,17,18]. The only structural protein not embedded in viral envelope, nucleocapsid (N) protein, is responsible for RNA-binding and RNA genome packaging to form helical ribonucleocapsid complexes in a beads-on-a-string arrangement for intracellular assembly [15], and is often selected as the target of antigen-based detection methods due to its abundance [12,19,20]. 

SARS-CoV-2 is best detected by respiratory samples, such as bronchioalveolar lavage [21], and nasopharyngeal swab (NPS) is one of the most commonly collected specimens for COVID-19 diagnosis. In this study, immunofluorescence using three different primary antibodies have been evaluated for their diagnostic performance in detecting various SARS-CoV-2 antigens in cells from clinical NPS. 

## 2. Materials and Methods

### 2.1. Clinical Specimens

Twenty-nine COVID-patients and 20 non-infected individuals were recruited in the Queen Mary Hospital, with clinical samples collected as NPS with cotton swabs that were immediately put in 2.5 mL of viral transport medium [22] and sent to Clinical Microbiology laboratory, Queen Mary Hospital of Hong Kong for viral investigation. Diagnoses were established using RT-PCR with LightMix Modular E-gene kit (TIB Molbiol, Berlin, Germany) [23] and confirmed by RdRp/Hel [24], with the leftover samples after confirmation were collected. Two clinical NPS from each patient, including NPS collected on admission and the day after admission, were used for performing immunofluorescence staining. Samples from non-infected individuals were proven negative for SARS-CoV-2, but some were infected with other respiratory viruses, including influenza A virus. The present study was approved by the Institutional Review Board of the University of Hong Kong/Hospital Authority Hong Kong West Cluster (UW13-372, version 20210517, 31 May 2021) and was performed in accordance with the guidelines and regulations.

### 2.2. Test Evaluation

Monoclonal antibody targeting single epitope of SARS-CoV N protein, cross-reacting with SARS-CoV-2 due to 91% homology in N proteins [25], was produced from infusing recombinant SARS-CoV N protein into BALB/c mice and utilizing the hybridoma technology, as described previously [26]. Polyclonal antibodies targeting SARS-CoV-2 N protein and RBD were produced as described [27], by direct injection of 30 µg of each recombinant proteins for 4 times, and retrieval of antisera from mice. 700 μL of aliquoted NPS samples were centrifuged at 6000 rpm for 1 min to remove the supernatant and collect the cell pellet, which was washed by resuspension in 700 μL of 1X phosphate-buffered saline (PBS) and centrifugation at 6000 rpm for 1 min. Cell pellet was resuspended in 50 μL of PBS for addition of 10 μL of cells in PBS to glass slides, followed by air drying and fixation by immersion in −20 °C chilled acetone in a −20 °C freezer for 10 min, allowing complete inactivation of viruses and permeabilization of cell membrane for antibodies to enter host cells [22]. The slides were air-dried and then stored in −80 °C until further usage. All steps completed before fixation with −20 °C chilled acetone were carried out within a class II biosafety cabinet for safety reasons. Indirect immunofluorescence staining of glass slides fixed with cells were done by sequential incubation with antibodies at 36 °C for 45 min, with the addition of 10 μL of primary antibody and 10 μL of 1:40 fluorescein isothiocyanate (FITC)-conjugated goat anti-mouse IgG antibodies (Millipore, Merck, Germany) in 0.0005% Evans blue (Sigma-Aldrich, Merck, Germany). After washing with 1X PBS, the slides were air-dried, and 1:1 glycerol-PBS was added to all spots to stabilize the antigen-antibody interactions, reduce photobleaching and allow better observation under the epifluorescence microscope with excitation at 490 nm and emission at 520 nm [28]. For each sample, all fields were screened using 100X magnification, and 200X magnification was used to further characterize all positive fluorescent signals to determine the positively stained cell type and distinguish from non-specific signals. Positive cells were identified as speckled pattern of apple-green fluorescent signals when staining N protein and membrane apple-green fluorescent signals when staining RBD, for which all cells in 3 fields of each sample were counted at 200X magnification to quantify the percentage of positive cells. 

### 2.3. Cells from Viral Culture

Smears of Vero E6 cells infected with SARS-CoV-2 were used as internal controls, including positive and conjugate control by addition of 10 μL of primary antibody and PBS, respectively, that was followed by FITC-conjugated secondary antibody; identification of non-specificity or cross-reactivity of normal mice antibodies to establish the usage of polyclonal antibodies for diagnosis, the addition of pooled antibodies from non-immunized mice as primary antibodies were used, followed by FITC-conjugated secondary antibody. Titers of primary antibodies for optimal visualization and differentiation between positive and negative cells were determined by serial dilution of primary antibody, which were 1:500, 1:100 and 1:100 dilution for SARS-CoV N protein, SARS-CoV-2 N protein and SARS-CoV-2 RBD, respectively.

To further illustrate the specificity of assay in order to identify any cross-reactivity with other viruses or cell lines, immunofluorescence was performed on 20 viruses, including SARS-CoV, MERS-CoV, HCoV-229E, HCoV-NL63, HCoV-OC43, influenza A virus, influenza B virus, respiratory syncytial virus, adenovirus, human metapneumovirus, parainfluenza virus types 1, 2, 3 and 4, human herpes virus 6 and 7, Japanese encephalitis virus, yellow fever virus, Zika virus and chikungunya virus; and 3 cell line controls including non-infected Vero, LLC-MK2 and Caco-2 cells. The cultured cells infected with SARS-CoV-2, SARS-CoV, MERS-CoV, yellow fever virus and chikungunya virus were prepared in a biosafety level 3 laboratory.

## 3. Results

Figure 1, Figure 2 and Figure 3 show samples that were identified as positive by immunofluorescence method using three primary antibodies, with each photo showing both positive and negative cells clearly delineated by the apple-green and red fluorescent signals. Quantity insufficient (QI) and non-specific signals (NS) were considered as invalid for results interpretation. Results of each specimen has been listed in Table 1. 

All NPS from recruited participants were used to perform RT-PCR for deducing the viral load in copies per mL from Ct values, in order to compare with immunofluorescence of the cells derived from the samples stained with the three primary antibodies. Among all recruited COVID-19 patients, the mean RT-PCR results of the NPS from the first two days after admission was 24.1 Ct value (interquartile range (IQR) = 28.4 − 18.9 = 9.5) and 1.08 × 10^9^ copies per mL (IQR = 9.34 × 10^8^ − 3.05 × 10^6^ = 9.31 × 10^8^. 

Median latencies of viral load in positive and non-positive results in indirect immunofluorescence were 9.40 × 10^8^ and 9.82 × 10^6^ for SARS-CoV N (Mann–Whitney U = 65, *p*-value < 0.05); 1.02 × 10^9^ and 8.13 × 10^6^ for SARS-CoV-2 N (Mann–Whitney U = 25, *p*-value < 0.05); 2.96 × 10^8^ and 2.81 × 10^7^ for SARS-CoV-2 RBD (Mann–Whitney U = 301, *p*-value = 0.061 ≥ 0.05). Statistically significant correlation between indirect immunofluorescence positivity and RT-PCR viral loads were demonstrated in immunofluorescence targeting N proteins but not RBD. Figure 4 revealed the relationship between percentages of positive cells in immunofluorescence and RT-PCR Ct values, which resonates with the results from Mann–Whitney U tests where significant correlation between viral load and immunofluorescence positivity was observed when targeting N protein but not RBD.

No correlation between RT-PCR results in Ct values categorized as <20, 20–24.99, 25–29.99 and ≥30 and sample quality in terms of number of respiratory epithelial cells categorized into QI and non-QI samples was demonstrated for primary antibodies against SARS-CoV N (χ^2^ (3, N = 58) = 3.12, *p*-value = 0.38 > 0.05), SARS-CoV-2 N (χ2 (3, N = 58) = 3.60, *p*-value = 0.31 > 0.05) and SARS-CoV-2 RBD (χ2 (3, N = 58) = 0.34, *p*-value = 0.95 > 0.05). As immunofluorescence results depend on the observation of respiratory epithelial cells to identify any SARS-CoV-2-infected cells, the quality of samples including the quantity and type of cells present may contribute to the absence of a clear trend between immunofluorescence and RT-PCR results in Figure 4.

Regarding specificity, cells from 20 non-infected controls were obtained in which 7 individuals had QI samples, among all other samples, negative results were obtained by antibodies targeting SARS-CoV and SARS-CoV-2 N proteins, but non-specificity of SARS-CoV-2 RBD antibody was identified as 6 NPS samples contained positive cells indistinguishable from true SARS-CoV-2-infected cells. For cultured virus-infected cells, all primary antibodies recognized SARS-CoV epitopes and caused positive results, revealing successful binding by antibody against SARS-CoV N and cross-reactivity by antibodies targeting SARS-CoV-2 N and RBD, which is reasonable due to high homology. Cross-reactivity with MERS-CoV by antibody against SARS-CoV-2 RBD was also recognized. All other tested virus-infected cells yielded negative results.

Since the interpretation of immunofluorescence requires a decent quality of clinical samples which means QI samples may impede the results, the 2 samples of each patient were collectively analyzed as a case rather than considering each sample separately. Utilizing the RT-PCR results as reference, the test performance parameters were calculated and shown in Table 2. The deduction of positive predictive value (PPV) and negative predictive value (NPV) depend on disease prevalence rate which was 42% among the tested samples.

## 4. Discussion

Immunofluorescence using polyclonal antibody targeting SARS-CoV-2 N protein demonstrated higher sensitivity than monoclonal antibody against SARS-CoV N protein, and higher specificity than polyclonal antibody against SARS-CoV-2 RBD, revealing that it would be the best option for staining of SARS-CoV-2-infected cells, with the sensitivity of 59% (95% CI 39–76%) and specificity of 98% (95% CI 87–100%). This yields a Youden (J) index of 0.56, as compared to Youden (J) indices of 0.46 and 0.39 in immunofluorescence targeting SARS-CoV N protein and SARS-CoV-2 RBD, respectively, which is lower than the acceptable threshold of 0.50. PPV and NPV were also highest in immunofluorescence targeting SARS-CoV-2 N protein. Only primary antibody targeting SARS-CoV-2 N protein should be considered for further application of immunofluorescence as a diagnostic test, resonating the choice of N protein as target in most antigen detection methods [12].

The lower sensitivity in immunofluorescence detecting SARS-CoV-2 infected cells by antibodies against SARS-CoV may be attributable to subtle antigenic differences and thus lower affinity of antigen–antibody interactions between SARS-CoV and SARS-CoV-2 despite high homology [29]. Another reason for the reduced sensitivity may be the monoclonality of the primary antibody against SARS-CoV N, although advantages of monoclonal antibodies include consistency and epitope specificity that would reduce batch-to-batch variations and enhance reproducibility [28]. On the contrary, polyclonal antibodies were chosen to target SARS-CoV-2 N and RBD given that the heterogeneity allows recognition of antigens using different epitopes, and hence enhanced binding effectiveness and minimal influences by subtle changes in protein conformation, such as due to environmental factors like pH and temperature, together with its relative simplicity and reduction in time, cost and efforts needed to generate the antibodies [30,31].

Regarding cross-reactivity and non-specificity, previous studies identified cross-reactivity of SARS-CoV-2 N protein with other coronaviruses, such as HCoV-229E [32,33] which was not demonstrated in this study, although cross-reactivity with HCoV-HKU1 was not tested due to unculturability in common cell lines, except in specific human airway epithelial cell lines [34,35]. Regarding polyclonal antibody targeting SARS-CoV-2 N, the only non-SARS-CoV-2-infected sample with positive immunofluorescence results was SARS-CoV-infected cells, yielding an appropriate specificity level, yet the specificity and PPV were relatively low due to the small sample size. Given the high homology between SARS-CoV and SARS-CoV-2 N protein, cross-reactivity between the two is expected and in fact, by excluding the cultured SARS-CoV-infected cells in deduction of test performance characteristics, both specificity and PPV would be 100%. The cross-reactivity with SARS-CoV could be considered an advantage as this allows simultaneous detection of both viruses. However, in addition to cross-reactivity with SARS-CoV due to 74% homology between SARS-CoV and SARS-CoV-2 RBD [36], polyclonal antibody against SARS-CoV-2 RBD demonstrated cross-reactivity with MERS-CoV and non-specificity in staining non-infected cells. This may be explained by its function of interacting with host cells via binding to multiple human receptors including angiotensin-converting enzyme 2 (ACE2), which is a common molecular present in various human cell types, as well as other molecules, such as dipeptidyl-peptidase 4 (DPP4), which is the receptor for MERS-CoV entry [37]. The role of RBD as a molecule for interaction, as compared to N protein located in viral envelope, may explain the poorer specificity. A potential controversy may be the polyclonality of antibody targeting SARS-CoV-2 RBD, yet polyclonal antibody against SARS-CoV-2 N protein serving as a comparison did not demonstrate such non-specificity, and non-immunized mice sera had no cross-reactivity or non-specificity, indicating that the low specificity of SARS-CoV-2 RBD antibody is unlikely due to polyclonality but rather a genuine difference in antigen–antibody binding specificity.

The presence of QI samples emphasizes the importance of proper sampling procedures carried out to obtain NPS, as QI samples contributed to the general poor sensitivity of immunofluorescence method as compared to other diagnostic methods. Moreover, the absence of statistically significant association between QI samples and viral load impinged on the identification of a clear relationship between Ct values against percentages of positive cells in IF, with QI samples being a significant external factor altering immunofluorescence results.

Previous publications have identified the sensitivity and specificity of other antigen-detection methods, including a meta-analysis of 29 studies on rapid antigen tests that indicate overall pooled sensitivity and specificity as 68% and 99% [38], and a recent large-scale study on a rapid antigen test (Roche/SD Biosensor) has shown similar diagnostic accuracy with 65% sensitivity and 100% specificity [39]. The immunofluorescence assay yields comparable sensitivity of 59% and specificity of 98%, owing to cross-reactivity with SARS-CoV. Moreover, test performance characteristics, including sensitivity, are not the only determinants of COVID-19 surveillance, but rather accessibility and frequency of testing, costs and response time should be prioritized [40,41], suggesting the importance of antigen-detection methods including in-house immunofluorescence as a simple way of detecting SARS-CoV-2-infected cells in both clinical specimens and cultured cells, which allows decentralized COVID-19 testing to enhance screening coverage [18] and alleviate material and manual shortages, for example, in large outbreaks that depleted available resources.

Benefits of indirect immunofluorescence include simplicity, cost-effectiveness, versatility allowing optimization of antibody targets as well as potential of staining in parallel yielding short turnaround time in large number of samples allows it as a competent method for the quick screening of clinical samples, or as a complementary testing method with other methods like RT-PCR. It allows the identification of intact virus-infected cells, which minimizes false positive results due to the persistence of viral components after patient recovery, as well as environmental contamination with viral components especially in areas with low COVID-19 prevalence [42,43,44]. This is in contrast with other methods detecting viral components that may be subjected to such external factors, such as RT-PCR, isothermal amplification, CRISPR-Cas12-based detection and next generation sequencing, in addition to generally being more expensive and complex. It also allows a much quicker way to confirm the presence of live virus when compared to viral culture, which is especially important for infection control precautions. Specific benefits of immunofluorescence over other antigen-detection methods, such as ELISA and rapid antigen tests, include the ability to visualize subcellular localizations of proteins, to assess quality in terms of checking type and number of cells present as well as ascertaining the presence of genuine positive signals rather than non-specific signals that permits detection of anomalies that may be missed by other antigen-detection methods [45]. Moreover, some shortcomings of immunofluorescence demonstrated in this study could be overcome by modifications of techniques, for example, streptavidin–biotin complex could enhance signal amplification and thus sensitivity, pooling of different specific monoclonal antibodies [28] to identify multiple epitopes of target antigen which increases probability of antigen detection or cytocentrifugation for smear preparation to concentrate cells and minimize QI samples. The time and effort required could be further shortened by the development of direct immunofluorescence by conjugating fluorophores to primary antibodies. Moreover, the plausibility of immunofluorescence as a diagnostic method was previously evaluated against many viruses, including SARS-CoV [46] and other coronaviruses, such as HCoV-229E and HCoV-OC43 [47], and is a traditional test for diagnosing many respiratory viruses, such as influenza viruses and human metapneumovirus [45,48,49]. Commercial respiratory virus panel for immunofluorescence include influenza A virus, influenza B virus, respiratory syncytial virus, adenovirus, human metapneumovirus and parainfluenza virus types 1, 2 and 3, which involve pooled monoclonal antibodies against antigens of common respiratory viruses, which are often used clinically for routine initial screening of respiratory samples [50,51]. This study reveals the potential of diagnosing SARS-CoV-2 infection by immunofluorescence, advocating for its incorporation into routine immunofluorescence respiratory virus panel for simultaneous respiratory virus detection, especially given the lack of cross-reactivity with all tested viruses except SARS-CoV. This allows for early identification and diagnosis of SARS-CoV-2 infection, with early implementation of infection control measures to prevent transmission of infection and early management of patients to ensure better prognosis with fewer long-term sequelae [2], thus enabling better surveillance, screening and diagnosis in the future, especially when there are only limited cases of COVID-19.

Nevertheless, disadvantages of immunofluorescence include subjectivity and inter-operator variability in results interpretation, as well as the reliance on intact virus-infected cells that may cause false negative results as viruses may not have infected human cells and viral antigens may not have been expressed or have already been degraded, in addition to the possibility of insufficient respiratory epithelial cells for results to be interpreted. Such limitations prevent accurate detection by immunofluorescence unlike other methods identifying free viruses, such as RT-PCR, and restricts potential sample types as certain specimens, such as deep-throat saliva, may not contain sufficient respiratory epithelial cells despite high load of free viruses as detected by other methods. Another issue would be the inability to deduce an accurate and practical lower limit of detection for detecting viruses in samples, given external factors including sample quality and antigenic expression in viral-infected respiratory epithelial cells. As compared to rapid antigen tests, immunofluorescence assays require extra equipment including a microscope and a dark room. The process of raising and purifying antibodies from mice may limit the application of immunofluorescence as an in-house test in hospitals or laboratories without animal room or protein-related equipment.

Some limitations of the study include the small sample size and potential sampling bias due to recruitment of hospitalized patients that may be more severe patients than in community settings. The collection day of clinical NPS were based on the admission date of patients rather than symptom onset day, leading to inevitable differences in infection stage between patients. Other unaccounted factors include sample quality and viral or host factors affecting cells in clinical NPS. In the future, expansion to include other specimen types, such as throat swab, and application in decision of hospital discharge of COVID-19 patients could be evaluated, together with investigating any changes in sensitivity when facing clinical samples of patients with SARS-CoV-2 variants. Direct comparison with rapid antigen tests could be done to further evaluate the potential utility of immunofluorescence as compared to rapid antigen tests. Protocol modifications to enhance sensitivity and development of more specific antibodies may be performed, together with development of automated machines for capturing of immunofluorescence microscopy photos, cell counting and quantification of fluorescent signal intensity to increase objectivity and efficiency in interpretation of results.

Ultimately, immunofluorescence could be used in combination with more complicated molecular assays, such as RT-PCR. It could be used as the first step of diagnosis or large-scale and routine screening by incorporating into simultaneous immunofluorescence respiratory virus panels to recognize any positive cases which would be isolated and managed immediately, followed by molecular confirmation of other samples identified as negative by antigen detection methods. The multiplex immunofluorescence allows SARS-CoV-2 detection to become a part of standard diagnostic procedures that allows testing even if the COVID-19 pandemic has ended, benefiting the community by early detection of cases and prevention of transmission due to asymptomatic spreaders, as well as optimizing resources and financial allocations.

## Figures and Tables

**Figure 1 diagnostics-11-02346-f001:**
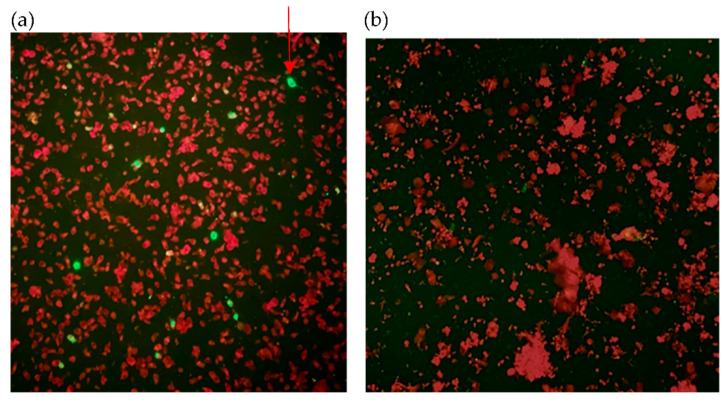
Smears of cells stained with primary antibody against SARS-CoV N protein (original magnification × 200). (**a**) Smear of cells from COVID-19 patients with positive results. (**b**) Smear of cells from non-infected individuals with negative results. Red arrow indicates positive cell.

**Figure 2 diagnostics-11-02346-f002:**
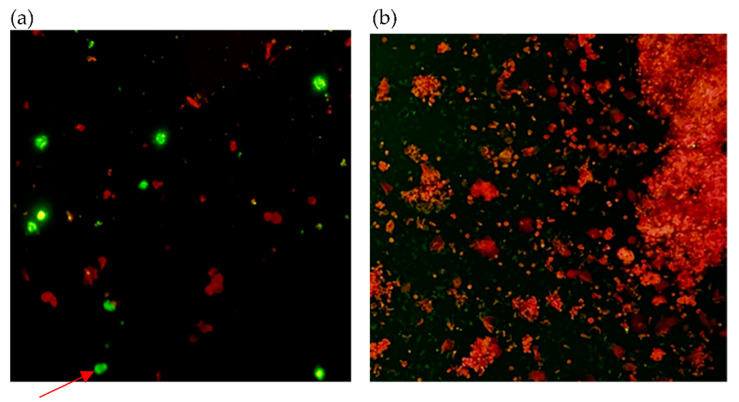
Smears of cells from clinical NPS from COVID-19 patients stained with primary antibody against SARS-CoV-2 N protein (original magnification × 200). (**a**) Smear of cells from COVID-19 patients with positive results. (**b**) Smear of cells from non-infected individuals with negative results. Red arrow indicates positive cell.

**Figure 3 diagnostics-11-02346-f003:**
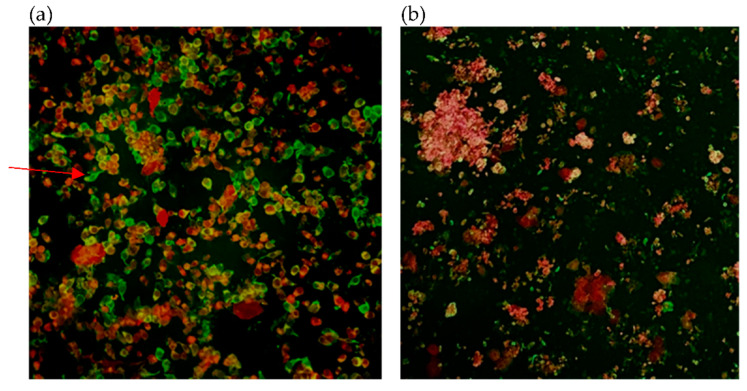
Smears of cells from clinical NPS from COVID-19 patients stained with primary antibody against SARS-CoV-2 RBD (original magnification × 200). (**a**) Smear of cells from COVID-19 patients with positive results. (**b**) Smear of cells from non-infected individuals with positive results indicating non-specificity. Red arrow indicates positive cell.

**Figure 4 diagnostics-11-02346-f004:**
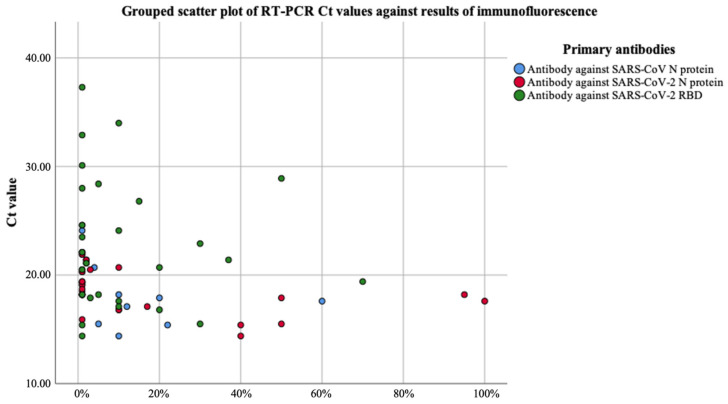
Grouped scatter plot of RT-PCR Ct values against results of indirect immunofluorescence.

**Table 1 diagnostics-11-02346-t001:** Detection of SARS-CoV-2 in cells from NPS samples of COVID-19 patients by indirect immunofluorescence using 3 primary antibodies.

Case Number	Collection Day	Result of Immunofluorescence			Result of RT-PCR	
		SARS-CoV N	SARS-CoV-2 N	SARS-CoV-2 RBD	Ct Value	Viral Load (copies/mL)
1	Day 1	+	+	QI	18.2	6.8 × 10^8^
1	Day 2	+	+	+	17.6	9.4 × 10^8^
2	Day 1	−	−	−	26.8	9.8 × 10^6^
2	Day 2	−	−	−	31.1	5.2 × 10^5^
3	Day 1	−	−	−	25.2	2.8 × 10^7^
3	Day 2	+	+	QI	19.3	1.5 × 10^9^
4	Day 1	−	−	+	30.1	9.8 × 10^5^
4	Day 2	QI	QI	QI	35.1	3.3 × 10^4^
5	Day 1	−	−	+	37.3	3.9 × 10^3^
5	Day 2	−	−	+	34.0	6.8 × 10^4^
6	Day 1	−	−	+	22.9	4.4 × 10^7^
6	Day 2	−	−	+	26.8	9.8 × 10^6^
7	Day 1	−	+	−	18.5	1.6 × 10^9^
7	Day 2	QI	QI	QI	18.7	1.2 × 10^9^
8	Day 1	+	+	QI	19.1	1.0 × 10^9^
8	Day 2	+	+	+	22.1	1.2 × 10^8^
9	Day 1	+	+	+	15.5	2.7 × 10^9^
9	Day 2	+	−	+	24.1	1.6 × 10^7^
10	Day 1	+	+	+	18.2	1.9 × 10^9^
10	Day 2	+	+	+	19.4	3.3 × 10^8^
11	Day 1	+	+	+	17.9	7.8 × 10^8^
11	Day 2	QI	QI	QI	23.0	4.1 × 10^7^
12	Day 1	−	+	+	24.6	7.0 × 10^7^
12	Day 2	−	−	+	28.0	8.1 × 10^6^
13	Day 1	+	+	+	20.5	1.0 × 10^9^
13	Day 2	+	+	+	20.7	2.9 × 10^9^
14	Day 1	+	+	+	14.39	2.3 × 10^10^
14	Day 2	QI	QI	QI	22.44	1.7 × 10^8^
15	Day 1	QI	QI	QI	20.9	6.1 × 10^7^
15	Day 2	QI	QI	QI	30.9	1.3 × 10^5^
16	Day 1	+	+	+	18.2	3.0 × 10^8^
16	Day 2	QI	QI	QI	23.0	2.4 × 10^7^
17	Day 1	−	+	QI	18.7	9.6 × 10^8^
17	Day 2	+	+	−	20.3	1.7 × 10^8^
18	Day 1	−	−	QI	37.6	1.9 × 10^3^
18	Day 2	−	−	−	30.7	3.1 × 10^5^
19	Day 1	−	−	−	20.9	6.1 × 10^7^
19	Day 2	−	+	−	15.9	1.2 × 10^9^
20	Day 1	−	−	NS	37.8	2.4 × 10^3^
20	Day 2	−	−	−	33.9	1.4 × 10^5^
21	Day 1	+	+	+	16.8	3.8 × 10^9^
21	Day 2	+	+	+	21.4	3.1 × 10^8^
22	Day 1	−	−	+	32.9	4.9 × 10^5^
22	Day 2	QI	−	QI	27.4	7.0 × 10^6^
23	Day 1	−	−	+	23.5	1.0 × 10^8^
23	Day 2	+	+	+	17.1	3.3 × 10^9^
24	Day 1	QI	QI	QI	28.9	2.8 × 10^6^
24	Day 2	QI	QI	QI	33.2	2.0 × 10^5^
25	Day 1	+	−	+	21.1	3.7 × 10^8^
25	Day 2	+	+	+	15.4	1.1 × 10^10^
26	Day 1	−	−	−	25.2	2.8 × 10^7^
26	Day 2	QI	QI	QI	28.4	3.9 × 10^6^
27	Day 1	−	−	+	28.9	1.2 × 10^6^
27	Day 2	−	−	+	28.4	1.7 × 10^6^
28	Day 1	−	−	−	24.2	3.3 × 10^7^
28	Day 2	−	−	−	25.3	1.6 × 10^7^
29	Day 1	+	+	−	18.9	9.0 × 10^8^
29	Day 2	−	QI	−	21.9	1.1 × 10^8^

**Table 2 diagnostics-11-02346-t002:** Test performance characteristics of immunofluorescence of the three primary antibodies.

Parameters	SARS-CoV N	SARS-CoV-2 N	SARS-CoV-2 RBD
Sensitivity [95% confidence interval (CI)]	48% (29–67%)	59% (39–76%)	59% (39–76%)
Specificity [95% confidence interval (CI)]	98% (87–100%)	98% (87–100%)	80% (64–91%)
Positive predictive value (PPV) [95% confidence interval (CI)]	93% (66–99%)	94% (71–99%)	68% (52–81%)
Negative predictive value (NPV) [95% confidence interval (CI)]	72% (65–79%)	76% (68–83%)	73% (63–81%)
Youden (J) index	0.46	0.56	0.39

## Data Availability

The data used to support the findings of this study are included within the article.

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
