# Peer review of "In-House Immunofluorescence Assay for Detection of SARS-CoV-2 Antigens in Cells from Nasopharyngeal Swabs as a Diagnostic Method for COVID-19"

_diagnostics, 2021, doi:10.3390/diagnostics11122346_

Round 1

Reviewer 1 Report

1. Immunofluorescence staining of infected cells from nasopharyngeal or throat swabs is a well established diagnostic tool in clinical virology which offers a rapid diagnostic method and can potentially be constructed for the simultaneous detection of multiple pathogens. The authors had used this simple methods to examine it's potential for clinical diagnosis of SARS-CoV-2 infection and presented data on their evaluation of different antibody preparations for specificity and sensitivity analysis.

2. Both the introduction and discussion can be substantially reduced and improved by maintaining focus on information relating to the current study.

3. Discussion on clinical use should be more elaborated such as the multiplexing the detection of different viruses such as influenza, RSV as well as other important coronaviruses and its impact on infection control as well as the initiation of antiviral treatment.

Author Response

Thank you so much for your valuable comments and suggestions.

Point 1: Immunofluorescence staining of infected cells from nasopharyngeal or throat swabs is a well established diagnostic tool in clinical virology which offers a rapid diagnostic method and can potentially be constructed for the simultaneous detection of multiple pathogens. The authors had used this simple methods to examine it's potential for clinical diagnosis of SARS-CoV-2 infection and presented data on their evaluation of different antibody preparations for specificity and sensitivity analysis.

 Response 1: Thank you for your valuable comments.

Point 2: Both the introduction and discussion can be substantially reduced and improved by maintaining focus on information relating to the current study.

Response 2: Some irrelevant information has now been removed from the introduction and discussion, although some specific additional information has been added in response to other reviewers’ comments.

Point 3: Discussion on clinical use should be more elaborated such as the multiplexing the detection of different viruses such as influenza, RSV as well as other important coronaviruses and its impact on infection control as well as the initiation of antiviral treatment.

 Response 3: The details of multiplex immunofluorescence respiratory virus panel have now been added, where antibodies against SARS-CoV-2 N protein could be added to allow routine screening of relevant viruses simultaneously. The routine screening of respiratory samples would allow early identification of patients with SARS-CoV-2-infected cells (indicative of live viruses) which permits early infection control and initiation of antiviral treatment, leading to better prevention of SARS-CoV-2 transmission and deterioration or development of complications in COVID-19 patients. These have been included in Lines 334-347.

Reviewer 2 Report

Lam and colleagues describe a study where immunofluorescence-based assay was used for SARS-CoV-2 detection in nasophatyngeal swabs. From what I understand, the antibodies tested were generated as part of the study, however the description of the antibody generation is very short without any references. Source of the proteins used for immunization is lacking.

The authors use generated antibodies for detection of SARS-CoV-2 positive samples in patients admitted to Queen Mary Hospital of Hong Kong. Results are compared against a RT-PCR based assay and show limited sensitivity and moderate-to-good specificity (98% for the best two antibodies). Obtained results should be taken with caution given a relatively small sample size. 

In the introduction authors focus on high cost, relative length and complexity of PCR based methods for SARS-CoV-2 detection. I agree with the first argument, but regarding the relative length and complexity perhaps it should be mentioned that molecular diagnostic solutions using all-in one systems where RNA extraction, reverse transcription, amplification and detection take place within the same device in under 2 hours are available, IVD certified, FDA approved and widely used in non-specialized labs.

I am a bit surprised not to see a broader description of lateral-flow antigen assays in the introduction given their universal use around the world and reasonably good sensitivity:specificity ratios given the low cost and ease of use. As these types of assays are even easier to perform than immunofluorescence (no need for a microscope) they should be described in more detail.

Indeed it would be interesting to compare immunofluorescence to rapid antigen tests. However, since the volume of many samples was insufficient to perform immunofluorescence with all three tested antibodies within the study, it is understandable that this would be difficult. Samples are probably already used anyway. Perhaps authors should compare the results obtained with other published data on rapid antigen tests specificity and sensitivity such as:

https://pubmed.ncbi.nlm.nih.gov/34242764/

and several others.

The authors mention potential application in community setting, however this would be rather difficult regarding the need for microscope and a dark room. In comparison rapid antigen tests can be performed much easier.

I would suggest rewriting the manuscript with focus on comparison (specificity and sensitivity) between the method described and widely used rapid antigen tests since they would be used in similar settings since authors emphasize the relatively low cost and simplicty of their approach.

Specific comments.

Line 21:

Evaluate the potential of indirect immunofluorescence as an in-21 house diagnostic method for SARS-CoV-2 antigens from nasopharyngeal swabs (NPS).

I guess the authors meant something in the line of: The goal of the research was to “…”

Line 25:

CoV-2 nucleocapsid protein and receptor-binding domain

“of SARS-CoV-2 pike protein” should be added

Lines 223 -226

Since the quality of samples prevent interpretation of immunofluorescence results of 223 all samples, the two samples of each participant were considered together as a case, con-224 tributing to the total number of 29 cases with different numbers of positive cases rather 225 than positive samples out of 58 samples which was largely impacted by QI samples.

Should be rewritten, difficult to follow as one sentence.

Lines 292-294

I think that claim of lowering the number of false positives compared to RT-PCR given the positive predictive value of 94% for the best antibody tested is unsubstantiated.

Figure 4 is difficult to read perhaps bar where several CT ranges are plotted against tested antibodies could be used instead.

Author Response

Thank you so much for your valuable comments and suggestions.

Point 1: From what I understand, the antibodies tested were generated as part of the study, however the description of the antibody generation is very short without any references. Source of the proteins used for immunization is lacking.

Response 1: This has been rewritten for clarity in Lines 103-108: “Monoclonal antibody targeting single epitope of SARS-CoV N protein, cross-reacting with SARS-CoV-2 due to 91% homology in N proteins [25], was produced from infusing re-combinant SARS-CoV N protein into BALB/c mice and utilizing the hybridoma technology as described previously [26]. Polyclonal antibodies targeting SARS-CoV-2 N protein and RBD were produced as described [27], by direct injection of 30 µg of each recombinant proteins for 4 times, and retrieval of antisera from mice.

 Point 2: The authors use generated antibodies for detection of SARS-CoV-2 positive samples in patients admitted to Queen Mary Hospital of Hong Kong. Results are compared against a RT-PCR based assay and show limited sensitivity and moderate-to-good specificity (98% for the best two antibodies). Obtained results should be taken with caution given a relatively small sample size.

Response 2: One of the limitations of the study would be the small sample size which has been discussed in Discussion section (Lines 364-366: “Some limitations of the study include the small sample size and potential sampling bias due to recruitment of hospitalized patients that may be more severe patients than in community setting.”)

 Point 3: In the introduction authors focus on high cost, relative length and complexity of PCR based methods for SARS-CoV-2 detection. I agree with the first argument, but regarding the relative length and complexity perhaps it should be mentioned that molecular diagnostic solutions using all-in one systems where RNA extraction, reverse transcription, amplification and detection take place within the same device in under 2 hours are available, IVD certified, FDA approved and widely used in non-specialized labs.

 Response 3: The disadvantages including high cost, relative length and complexity of PCR-based methods were referring to traditional methods in conventional laboratory. The all-in-one sample-to-result PCR-based methods is, however, highly costly per reaction. The distinction between conventional RT-PCR and all-in-one PCR-based method has now been clarified in the Introduction (Lines 49-55: “Reverse transcription polymerase chain reaction (RT-PCR) is the current gold-standard for SARS-CoV-2 detection [3], which is highly sensitive but require designated thermocyclers and specific expertise in molecular techniques, high expenses and generally long turn-around time due to complicated procedures, although sample-to-result PCR-based methods have been developed for use in non-specialized laboratories to simplify and shorten the process with the cost per reaction being high.”)

Point 4: I am a bit surprised not to see a broader description of lateral-flow antigen assays in the introduction given their universal use around the world and reasonably good sensitivity:specificity ratios given the low cost and ease of use. As these types of assays are even easier to perform than immunofluorescence (no need for a microscope) they should be described in more detail.

Response 4: The description of lateral-flow antigen assays has now been included in introduction (Lines 66-69: “various lateral-flow rapid antigen tests utilizing immunochromatographic assay or fluorescence immunoassay are widely used as point-of-care tests given their simplicity and low technical requirements, although accuracy and utility of rapid antigen tests have been reported to be variable [12,13].”).

Point 5: Indeed it would be interesting to compare immunofluorescence to rapid antigen tests. However, since the volume of many samples was insufficient to perform immunofluorescence with all three tested antibodies within the study, it is understandable that this would be difficult. Samples are probably already used anyway. Perhaps authors should compare the results obtained with other published data on rapid antigen tests specificity and sensitivity such as: https://pubmed.ncbi.nlm.nih.gov/34242764/ and several others.

 Response 5: As insufficient volume of samples was available for testing using rapid antigen tests, direct comparison with rapid antigen tests have not been performed. A meta-analysis of 29 studies on rapid antigen tests, and the quoted study has now been used for comparison with the sensitivity and specificity of the immunofluorescence assay (Lines 292-298: “Previous publications have identified sensitivity and specificity of other antigen-detection methods, including a meta-analysis of 29 studies on rapid antigen tests that indicate overall pooled sensitivity and specificity as 68% and 99% [38], and a recent large-scale study on a rapid antigen test (Roche/SD Biosensor) has shown similar diagnostic accuracy with 65% sensitivity and 100% specificity [39]. The immunofluorescence assay yields comparable sensitivity of 59%, and specificity of 98% owing to cross-reactivity with SARS-CoV.”).

The comparison between immunofluorescence and rapid antigen tests have now been included as a further investigation (Lines 329-334), together with protocol modifications including direct immunofluorescence by conjugation of fluorophores to primary antibody and use of pooled monoclonal antibodies instead of polyclonal antibodies.

 Point 6: The authors mention potential application in community setting, however this would be rather difficult regarding the need for microscope and a dark room. In comparison rapid antigen tests can be performed much easier.

Response 6: A limitation of application of immunofluorescence would be the equipment needed which is now addressed in Lines 359-363. The application in other settings has now been clarified to be the decision of when to discharge COVID-19 patients from hospital, as persistence of viral components in recovered patients may render other methods including RT-PCR and rapid antigen tests less useful in determining infectivity, unlike immunofluorescence that detects virus-infected cells indicative of live viruses (Lines 316-318).

 Point 7: I would suggest rewriting the manuscript with focus on comparison (specificity and sensitivity) between the method described and widely used rapid antigen tests since they would be used in similar settings since authors emphasize the relatively low cost and simplicty of their approach.

Response 7: It is true that immunofluorescence may be better compared with rapid antigen tests given the similar advantages of low cost and simplicity, but as mentioned in response 5, the insufficient sample volume has prevented direct comparison of immunofluorescence and rapid antigen tests, thus only comparison with the gold-standard of RT-PCR has been performed in the study. Yet, published data of rapid antigen tests have now been utilized for comparison (Lines 292-296), and specific benefits of immunofluorescence over other antigen-based detection methods including rapid antigen tests have been emphasized (Lines 318-323: “Specific benefits of immunofluorescence over other antigen-detection methods, such as ELISA and rapid antigen tests, include ability to visualize subcellular localizations of proteins, to assess quality in terms of checking type and number of cells present, as well as ascertaining presence of genuine positive signals rather than non-specific signals that permits detection of anomalies that may be missed by other antigen-detection methods [45].”)

 Point 8: Line 21: Evaluate the potential of indirect immunofluorescence as an in-21 house diagnostic method for SARS-CoV-2 antigens from nasopharyngeal swabs (NPS). I guess the authors meant something in the line of: The goal of the research was to “…”

 Response 8: This issue has now been corrected: This study aims to evaluate (Line 21).

 Point 9: Line 25: CoV-2 nucleocapsid protein and receptor-binding domain “of SARS-CoV-2 spike protein” should be added

Response 9: This issue has now been corrected (Line 25).

 Point 10: Lines 223 -226: “Since the quality of samples prevent interpretation of immunofluorescence results of 223 all samples, the two samples of each participant were considered together as a case, con-224 tributing to the total number of 29 cases with different numbers of positive cases rather 225 than positive samples out of 58 samples which was largely impacted by QI samples.” Should be rewritten, difficult to follow as one sentence.

 Response 10: This sentence has now been rewritten for clarity (Lines 226-228: “Since interpretation of immunofluorescence requires a decent quality of clinical samples which means QI samples may impede the results, the 2 samples of each patient were collectively analysed as a case rather than considering each sample separately.”).

Point 11: Lines 292-294: I think that claim of lowering the number of false positives compared to RT-PCR given the positive predictive value of 94% for the best antibody tested is unsubstantiated.

 Response 11: As immunofluorescence detects only antigens in virus-infected cells, a positive immunofluorescence would signify that the infection of cells was genuinely active, rather than the persistence of viral components including RNA after patient recovery or the contamination of viral components that may result in positive results in RT-PCR. The basis behind the claim of having a lower false positive was the principle of immunofluorescence that only detects virus-infected cells that suggest the presence of live viruses that are infective (Lines 306-313, 316-318).

Regarding the positive predictive value, the low PPV was due to the small sample size because the PPV was in fact calculated from a total of 15 test-positive cases which included 14 true positive cases and 1 false positive case which was in fact Vero E6 cells infected with SARS-CoV, indicating only cross-reactivity with SARS-CoV but none of the other tested samples including 20 non-SARS-CoV-2-infected clinical samples, cultured cells infected with SARS-CoV, MERS-CoV, HCoV-229E, HCoV-NL63, HCoV-OC43, influenza A virus, influenza B virus, respiratory syncytial virus, adenovirus, human metapneumovirus, parainfluenza virus types 1, 2, 3 and 4, human herpes virus 6 and 7, Japanese encephalitis virus, yellow fever virus, Zika virus and chikungunya virus, and cell line controls including non-infected Vero, LLC-MK2 and Caco-2 cells.

As SARS-CoV belongs to the same family as SARS-CoV-2, it would be anticipated for cross-reactivity to occur, with none of the other tested viruses having cross-reactivity, which could serve as an advantage of the assay by simultaneous detection of both SARS-CoV and SARS-CoV-2. If the SARS-CoV-infected cells were excluded in the calculation, both specificity and PPV would be 100%. It would also not be a problem for inability to differentiate between SARS-CoV and SARS-CoV-2 given that SARS-CoV is no longer circulating in the community, thus a positive immunofluorescence results targeting SARS-CoV-2 N protein would confidently suggest true presence of SARS-CoV-2-infected cells.

This has now been addressed in the Discussion (Lines 263-272).

Point 12: Figure 4 is difficult to read perhaps bar where several CT ranges are plotted against tested antibodies could be used instead.

Response 12: As the original goal of Figure 4 (P.7) was to compare the RT-PCR Ct value with the immunofluorescence percentage of positive cells (counted on microscope), it would be difficult to draw if a scatter plot is not used. As we have noticed that a major reason for the graph to be incomprehensible would be the cluster of datapoints with 0% positive cells which correlated with negative immunofluorescence results, the scatter plot has now been redrawn without including the negative results for the purpose of clarity.

Reviewer 3 Report

  1. What is the author opinion on comparative analysis of presented Immunofluorescence Assay methodology with traditionally used RT-qPCR methodology or RTPCR/MALDITOF; Isothermal amplification (OMEGA amplification); Next generation gene sequencing and CRISPR-Cas12 techniques that were introduced for the detection of SARS-CoV-2.
  2. An advantage of present method over the other developed techniques may be highlighted in the revised work.
  3. Concentration of polyclonal antibody utilised may be pointed
  4. Fluorescence intensities (ɸf values) should be measured and pointed.
  5. The authors should obtain and added a photophysical properties (ex, em, quantum yield) concerning the fluorescence by polyclonal antibody utilised.
  6. Scale to images in figure 1-3 is missing.

Author Response

Thank you so much for your valuable comments and suggestions.

Point 1: What is the author opinion on comparative analysis of presented Immunofluorescence Assay methodology with traditionally used RT-qPCR methodology or RTPCR/MALDITOF; Isothermal amplification (OMEGA amplification); Next generation gene sequencing and CRISPR-Cas12 techniques that were introduced for the detection of SARS-CoV-2.

 Response 1: The benefits of immunofluorescence assay has been described in Lines 306-323 and the disadvantages in Lines 348-363.

 All of the listed methodologies involve detection of viral components, which may be subjected to the persistence of viral components after patient recovery or the environmental contamination by viral components, as compared to immunofluorescence which could only detect virus-infected cells that indicate presence of live viruses. Moreover, the stated methods are generally more expensive, and may be more complex in terms of procedures such as next generation sequencing.  

Point 2: An advantage of present method over the other developed techniques may be highlighted in the revised work.

Response 2: The benefits of immunofluorescence assay over other developed techniques have been described in Lines 306-323, and particularly benefits over other antigen-detection methods in Lines 318-323. The major advantage would be the ability to identify virus-infected cells indicative of live viruses, as opposed to other methods which may be affected by persistence of viral components after recovery, or environmental contamination. Other advantages include simplicity, cost-effectiveness, versatility and short turnaround time.

 Point 3: Concentration of polyclonal antibody utilised may be pointed

 Response 3: By serial dilution of the three primary antibodies, the titers for optimal visualization and differentiation between positive and negative cells were deduced, which were 1:500, 1:100 and 1:100 dilution for SARS-CoV N protein, SARS-CoV-2 N protein and SARS-CoV-2 RBD respectively (Lines 139-142).                                                       

Point 4: Fluorescence intensities (ɸf values) should be measured and pointed.

Response 4: Slides were viewed at a magnification of 100X or 200X under epifluorescence illumination using FITC filter of a fluorescence microscope. The system is unable to quantify fluorescence intensity. We clarify: “allow better observation under the epifluorescence microscope with excitation at 490 nm and emission at 520 nm” (Lines 123-124). The positive fluorescent cell has been pointed out in figures.

 Point 5: The authors should obtain and added a photophysical properties (ex, em, quantum yield) concerning the fluorescence by polyclonal antibody utilised.

 Response 5: Please see “Response 4”.

Point 6: Scale to images in figure 1-3 is missing.

Response 6: Magnification has been added in figures (P4-5).

Reviewer 4 Report

This manuscript is well written and organized. Authors describe a detection method for COVID-19 by multiplex immunofluorescence, targeting SARS-CoV N protein, SARS-CoV2 N, and SARS-CoV2 RBD. My comments are as follows.

  1. In Abstract, authors claim that “Polyclonal antibody against SARS-CoV2 N protein had the highest sensitivity and specificity……, detecting 17 out of 29 RT-PCR-confirmed COVID-19 cases……”
    - Is this approach appropriate for application of large-scale and routine diagnostic of respiratory viruses including SARS-CoV due to its polyclonal antibody against SARS-CoV2-N has the highest sensitivity and specificity of 58.6% positive among 29 confirmed cases?
  2. In introduction, “Direct viral isolation by electron microscopy or culture in cell…… application as a routine testing method [5-7]”, “Spike (S) proteins are large…… are triggered by receptor-binding of S1 subunit [14-16]”, and “Predominantly affecting the respiratory tract, ……collected specimens for COVID-19 diagnosis.”
    - It will be easier to understand if sentences are shortened or split.
  3. Twenty samples from non-infected individuals are included in this study, are they healthy donors? Is there any samples coinfected with other respiratory viruses?
  4. In 2.2 Test evaluation, SARS-CoV N protein has 91% homology with SARS-CoV2, how about the RBD region?
  5. The brand information of reagents is missing in 2. Materials and Methods part.
  6. Cells have low level of fluorescent signal naturally, how to distinguish the real positive signal from background noise or avoid false positive from background noise?
  7. Figure 4 legend need to be extended.
  8. Line 228, add the full names of “PPV” and “NPV”.
  9. A recent publication (https://doi.org/10.1073/pnas.2105968118) shows reverse-transcribed SARS-CoV-2 RNA can integrate into genome of cultured human cells. Is there any possibility to have false negative in this study?
  10. Any data from oral/throat swabs?
  11. The duration and process to raise and purification of monoclonal/polyclonal antibodies from mice will limit the use of this method in hospital and general lab which doesn’t have animal room and protein related equipment.

Author Response

Thank you so much for your valuable comments and suggestions.

Point 1: In Abstract, authors claim that “Polyclonal antibody against SARS-CoV2 N protein had the highest sensitivity and specificity……, detecting 17 out of 29 RT-PCR-confirmed COVID-19 cases……”

- Is this approach appropriate for application of large-scale and routine diagnostic of respiratory viruses including SARS-CoV due to its polyclonal antibody against SARS-CoV2-N has the highest sensitivity and specificity of 58.6% positive among 29 confirmed cases?

 Response 1: We would like to clarify that polyclonal antibody against SARS-CoV-2 N protein had the highest sensitivity and specificity among the three antibodies tested in the study (Lines 28-29). The sentence has been modified: Line 36 “thus allowing additional routine testing for diagnosis and surveillance of SARS-CoV-2”

Point 2: In introduction, “Direct viral isolation by electron microscopy or culture in cell…… application as a routine testing method [5-7]”, “Spike (S) proteins are large…… are triggered by receptor-binding of S1 subunit [14-16]”, and “Predominantly affecting the respiratory tract, ……collected specimens for COVID-19 diagnosis.”

- It will be easier to understand if sentences are shortened or split.

Response 2: These three sentences have now been shortened for ease of comprehension (Lines 56-62, Lines 73-77, and Lines 82-84).

Point 3: Twenty samples from non-infected individuals are included in this study, are they healthy donors? Is there any samples coinfected with other respiratory viruses?

 Response 3: The non-infected individuals refer to anyone proven negative for SARS-CoV-2, which included healthy individuals and patients infected with other respiratory viruses such as influenza A virus (Lines 97-98).

Point 4: In 2.2 Test evaluation, SARS-CoV N protein has 91% homology with SARS-CoV2, how about the RBD region?

 Response 4: The homology between SARS-CoV and SARS-CoV-2 RBD was reported to be 74%, which has now been added (Lines 271-272).

Point 5: The brand information of reagents is missing in 2. Materials and Methods part.

Response 5: It has now been corrected (Lines 120-121).

Point 6: Cells have low level of fluorescent signal naturally, how to distinguish the real positive signal from background noise or avoid false positive from background noise?

 Response 6: Evans blue was added in FITC conjugate (Lines 120-121) which stained negative cells appearing red under fluorescence microscope. This allows good contrast between positive cells (apple green) and negative cells (red).  

Point 7: Figure 4 legend need to be extended.

 Response 7: The legend of Figure 4 (P.7) has now been extended.

Point 8: Line 228, add the full names of “PPV” and “NPV”.

Response 8: This has now been added (Line 230).

Point 9: A recent publication (https://doi.org/10.1073/pnas.2105968118) shows reverse-transcribed SARS-CoV-2 RNA can integrate into genome of cultured human cells. Is there any possibility to have false negative in this study?

 Response 9: We are uncertain whether reverse-transcribed SARS-CoV-2 RNA can integrate into genome of human cells in vivo or not. In our study, all patient samples positive in immunofluorescence against SARS-CoV-2 N protein are SARS-CoV-2 RT-PCR positive, while all tested SARS-CoV-2 RT-PCR negative patient samples are negative in immunofluorescence against SARS-CoV-2 N protein. It seems to be the immunofluorescence test against SARS-CoV-2 N has good specificity. If reverse-transcribed SARS-CoV-2 RNA can integrate into genome of human cells in vivo, it is likely RT-PCR test for SARS-CoV-2 would be positive but immunofluorescence would likely negative. It would only be positive by immunofluorescence if genetic materials from SARS-CoV-2 are expressed and transcribed.

Point 10: Any data from oral/throat swabs?

 Response 10: This has not been performed as only nasopharyngeal swabs were collected from the participants, yet this has now been included as a further investigation (Line 369-372). 

Point 11: The duration and process to raise and purification of monoclonal/polyclonal antibodies from mice will limit the use of this method in hospital and general lab which doesn’t have animal room and protein related equipment.

Response 11: This would be a limitation of the immunofluorescence assay as an in-house test in hospitals and general laboratories without such equipment (Lines 359-363); yet, this could possibly be overcome by the incorporation of antibody against SARS-CoV-2 antigens into multiplex immunofluorescence panels for respiratory viruses that are commercially available, which makes it much more convenient and accessible (Lines 334-347).

Round 2

Reviewer 2 Report

I appreciate the effort of the authors in clarifying the discussed points.